# Effects of a Physical Exercise Programme through Service-Learning Methodology on Physical Activity, Physical Fitness and Perception of Physical Fitness and Health in University Students from Spain: A Preliminary Study

**DOI:** 10.3390/ijerph20043377

**Published:** 2023-02-15

**Authors:** Antonio Jesús Casimiro-Andújar, Eva Artés-Rodríguez, David M. Díez-Fernández, María-Jesús Lirola

**Affiliations:** 1Department of Education, Faculty of Education Sciences, University of Almería, 04120 Almería, Spain; 2Sport Research Group (CTS-1024), CERNEP Research Center, University of Almería, 04120 Almería, Spain; 3Area of Statistics and Operative Research, Department of Mathematics, Faculty of Sciences, University of Almería, 04120 Almería, Spain; 4Department of Psychology, Faculty of Psychology, University of Almería, 04120 Almería, Spain

**Keywords:** health promotion, UAL Activa, sedentary lifestyle, healthy habits

## Abstract

The practice of physical activity has been reported on countless occasions for the benefits it has on people’s holistic health. However, today’s society has high levels of inactivity and sedentary lifestyles, which highlights the importance of promoting active and healthy states in the population. As a mechanism to improve body composition, physical condition and perceived values of one’s own physical condition and health status, the implementation of a strength training programme in the university community was proposed using a methodology based on Service-Learning. The participants were 12 students as coaches and 57 students from different university degrees as coachees (17 boys and 40 girls); the ages of the participants ranged from 18 to 33 years (*M* = 22.00; *SD* = 2.96). The variables of body composition, physical fitness, physical activity level and perception of fitness and health were assessed. Differences between pre- and post-intervention results were analysed using the Student’s t-test and Wilcoxon test for ordinal self-perception variables. The results showed significant improvements in all the variables evaluated after the intervention. In conclusion, we would like to highlight the benefits of physical activity and the need to continue implementing action and intervention plans to encourage and promote its practice in all sectors of the population.

## 1. Introduction

Several studies reveal that a low level of physical activity (PA) or a sedentary lifestyle is associated with the development of various chronic [1], cardiovascular [2,3], metabolic [4,5] and musculoskeletal [6] diseases. Sedentary people are reported to account for 41% of the Spanish population over 18 years of age, and 60% of women over 15 years of age do not engage in leisure-time PA any day of the week [7]. Only 24% of university students are engaged in PA every day [8]. In line with a previous work, Castillo and Giménez [9] highlighted that 42% of university students practise PA once a week, although it reduced to 24% for a frequency of at least three days a week. It is also important to differentiate the intensity of physical activity, as it has been shown that different intensities of exercise (e.g., walking, running, interval training) lead to different fitness consequences for both over- and under-training [10,11].

University students also have low levels of physical fitness; this refers to the state of an individual’s body. Those who are in good physical condition are able to perform various activities effectively and vigorously, avoiding injury and with reduced energy expenditure. People who are in poor physical condition, on the other hand, feel tired soon after starting work, experiencing a progressive deterioration of their capacity and effectiveness. In this study, we specifically evaluated physical fitness in hand grip strength, lower body explosive strength, anterior trunk flexibility [12,13] and ischiosural flexibility [12], and general flexibility [14]. In addition, cardiovascular endurance and quadriceps strength endurance, which were not included previously [14], are included in the current study. Pang [15] highlighted advantages between university students who performed PA and those who did not, finding improvements in vital capacity and maximal oxygen consumption. 

The National Institute of Statistics from Spain shows that 61% of the population over 18 years of age is overweight or obese [7]. In addition to low levels of PA and physical fitness, several studies have shown that the prevalence of overweight and obesity in university students ranges from 17% to 38% [16,17,18,19,20,21]. Therefore, it is important to promote healthy physical activity habits that favour an adequate body composition, since a high fat percentage is associated with a higher frequency of metabolic diseases and comorbidities [22,23].

Another variable to be taken into account in terms of health promotion is not only the absence of disease but also to promote adequate levels of self-perception of health and physical condition, as recommended by the WHO [24], which promotes studies that take into account not only objective measures of health but also subjective measures such as the self-perceptions mentioned above. These have been shown to be improved thanks to the regular practice of physical activity, favouring that these self-perceptions take on a positive meaning [25].

In the university context, the service-learning methodology (SL) has been chosen worldwide as a didactic tool that enables links with society and integration with the theoretical knowledge taught in the classroom [26,27]. Several authors highlight that SL practices in students allow them to develop and put into play social skills and attitudes in real life [28,29]. In this way, SL responds to countless challenges that arise in today’s society and in the new European Higher Education Framework, promoting one of the basic functions of the university: the training of critical, active and responsible citizens [28,30]. That is why this project takes its basis from an SL methodology where curricular learning objectives are combined with those of service to the university community, in order to improve the realities of healthy and active habits of the participants [31].

The data reported highlight the importance of finding interventions to improve all these parameters in the university population. Physical exercise could be an effective alternative. Several studies have recorded the level of PA in university students [9,20,32,33]); have evaluated the level of physical fitness [12,13,34,35]; and have analysed the influence of PA on body composition in cross-sectional studies [21,36], on the inhibition of automatic responses [37], and on self-efficacy [38], but, to our knowledge, no study has evaluated the effect of a training programme. Therefore, filling the literature gap and meeting the demand raised by Barranco-Ruiz et al. [39] as a necessary line of work, the aim of this study was to analyse the effects of a physical exercise programme on levels of physical activity, physical fitness, body composition and the perception of physical fitness and health in university students through the application of a service-learning methodology. To this end, the following hypotheses were put forward in this study:

**Hypothesis** **H1.**
*An improvement in body composition by increasing lean mass and reducing body fat indexes.*


**Hypothesis** **H2.**
*A significant increase in the parameters on the evaluation of physical condition.*


**Hypothesis** **H3.**
*An increase in the number of days of high-intensity PA per week.*


**Hypothesis** **H4.**
*A positive improvement in the participants’ self-perception of their Physical Fitness and Health.*


## 2. Method

### 2.1. Study Design

A quasi-experimental study was conducted to assess the effect of a physical exercise programme on physical activity, physical fitness and body composition in university students. Two evaluations (pre- and post-intervention of 10 weeks of training) were conducted. 

### 2.2. Ethical Aspects

After being informed of the study objectives and protocol, conditions of participation, risks, and benefits and instructions of the training programme, participants were informed that they could leave the study at any time and signed an informed consent form approved by the ethics committee of the University of Almeria. 

### 2.3. Participants

The sample consisted of 12 students of the 3rd year of Physical Activity and Sport Sciences who acted as Coach involved in guiding and supervising the training programme, and 57 university students (17 boys and 40 girls) belonging to different degrees who received the mentioned programme, all of them from the University of Almeria. Participants ranged in age from 18 to 33 years (*M*_age_ = 22.00; *SD* = 2.96). As inclusion criteria, both groups of students (i.e., coaches and coachees) completed the Par-Q questionnaire, which indicated the need for medical consultation prior to starting the exercise programme (Canadian Society for Exercise Physiology) through a simple 7-question battery, with only one positive response requiring medical consultation [40]. In addition, a specific requirement as an inclusion criterion for coachees was to have a training attendance rate of at least 80%.

### 2.4. Protocol

The development of the programme took place in the facilities of the University of Almeria. The initial assessments were carried out on a total of 2 days, in the morning and afternoon, thus dividing the sample. The same happened with the post-intervention assessments. There was a training session before the start of the intervention that served to familiarise the participants with the training proposal, with the sports centre where it was carried out and with the nature of the effort. This methodology was used to control the intensity of the exercises in each session. Two experienced researchers were in charge of designing the training protocol, carrying out the evaluations and data collection, as well as accompanying the participants in the familiarisation session. The physical programme was carried out and controlled by volunteer students of the subject PA programmes for health in the 3rd year of the Degree in Physical Activity and Sport Sciences. 

### 2.5. Intervention Programme

The training programme, based on the Service-Learning (SL) methodology, with university students, focused mainly on muscle strengthening (MS) sessions, in accordance with the PA recommendations of the Junta de Andalucía [41]. It consisted of 10 weeks, two face-to-face sessions per week of 60 min of personal training (coach-coachee) on non-consecutive days, plus 1 scheduled training session). The sessions were designed with an initial part with a warm-up consisting of joint mobility, active stretching and 5 min of organic activation through continuous running on a treadmill; a main part of MS with a methodological proposal based on 3 circuits using auto-loads, elastic bands, weight machines, TRX or free weights, combined with aerobic exercise distributed in 3 intervals of 10’ after each lap of the circuit; and a final part of cool-down with stretching, looseness and breathing.

The main part of the 2 sessions in each of the 10 weeks, as well as the periodisation throughout the intervention, is detailed below, bearing in mind that a third day of cardiometabolic training is planned between 45′ and 90′ of moderate to vigorous intensity that the coachee practices on their own throughout the week: Week 1: self-loads. 10 exercises, 3 sets, 12 (20) repetitions ^1^, plus 30′ of aerobic work at 40% of the reserve HR.Week 2: elastic bands. 10 exercises, 3 sets, 12 (18) repetitions, plus 30′ of aerobic work at 45% HR Reserve.Week 3: weight machines 10 exercises, 3 sets, 10 (16) repetitions, plus 30′ of aerobic work at 50% FC Reserve.Week 4; weight machines. 10 exercises, 3 sets, 8 (14) repetitions, plus 30′ of aerobic work at 55% HR Reserve.Week 5: weight machines. 10 exercises, 3 sets, 6 (10) repetitions, plus 30′ of aerobic work at 60% HR Reserve.Week 6: suspension training with TRX. 10 exercises, 3 sets, 12 (18) repetitions, plus 30′ of aerobic work at 65% HR Reserve.Week 7: suspension training with TRX. 10 exercises, 3 sets, 10 (16) repetitions, plus 30′ of aerobic work at 70% HR Reserve.Week 8: free weights. 10 exercises, 3 sets, 10 (16) repetitions, plus 30′ of aerobic work at 75% HR Reserve.Week 9: free weights. 10 exercises, 3 sets, 8 (14) repetitions, plus 30′ of aerobic work at 80% HR Reserve.Week 10: free weights. 10 exercises, 3 sets, 6 (10) repetitions, plus 30′ of aerobic work at 85% HR Reserve.

^1^ 12 (20) means that he performs 12 repetitions but could do 8 more, in total 20; therefore, it does not end up causing excessive fatigue as the load is moderate.

### 2.6. Instruments 

Physical activity: It has been used the short version of the International Physical Activity Questionnaire (IPAQ). This questionnaire is composed of 7 questions that refer to the time spent being active during the previous 7 days. The short version of the IPAQ asks about 3 types of activity: walking, moderate activity and vigorous activity [42]. 

General physical fitness: It has been assessed through the tests belonging to the AFISAL-INEFC battery [43]. This battery has been previously used to assess the physical fitness of a Spanish university population [13]. The following tests were performed within the exercises of the battery: (1) maximum hand grip strength, evaluated with a digital dynamometer (Model T.K.K.540; Takei Scientific Instruments Co, Ltd., Yashiroda Akiha-Ku Niigata City, Japan), (2) abdominal strength-resistance (in this case by means of a frontal isometric plank with 4 supports, trying to reach the maximum time), (3) trunk flexibility, evaluated with a sit and reach test, and (4) lower body explosive strength (in this case by means of a horizontal jump with both feet together, trying to reach the maximum distance). Each test was supervised by two experienced researchers supported by students specialised in Physical Activity and Sport Sciences who were previously trained to guarantee a unification in the procedure and criteria when carrying out the tests.

Body composition: It has been evaluated through InBody720. Electrical impedance analysis using 4 different frequencies. This instrument uses eight electrodes, four in contact with the palm and thumb of both hands and four on the front and back of the soles of both feet. The subject stands with the sole in contact with the foot electrodes and grasps the hand electrodes. The data recorded are lean weight, % fat and visceral fat [44]. Test participants were required to attend early in the morning on an empty stomach, with the same conditions for all participants. 

Perception of physical fitness and health status: It was assessed by means of two direct questions taken from the questionnaire administered by Casimiro [45], whose wording for physical fitness was “How do you think your physical fitness is?” and for health status “How do you think your health is?”, both with five possible answers, *very bad, bad, normal, good, and very good*, categorised from 1 to 5, respectively.

### 2.7. Statistical Analysis

Descriptive statistics were analysed for the sample (*M* = Mean; *SD* = Standard Deviation), including age and gender. Subsequently, a bivariate correlation analysis was performed between the variables under study (i.e., body composition, physical fitness and physical activity level variables) in order to show evidence of the proper functioning of the assessment instruments used, thus providing insight into the behaviour of the related variables, as well as providing a sense of understanding of the phenomenon to which they refer and indicating irregularities, if any. Next, in order to analyse the possible significant differences between the data obtained in the pre-test and post-test, the Kolmogorov-Smirnov test was performed to reveal the normality of the sample. Finally, different mean comparison analyses were performed using Student’s *t*-test for related samples, with 95% confidence interval, and the effect size was calculated using Cohen’s *d* [46], where the effect can be categorised as small (0.2), medium (0.5) or large (0.8). For ordinal variables on perceived fitness and health status, the Wilcoxon test and its degree of significance (*p* < 0.05) were used. For data analysis and processing, SPSS statistical software version 24 was used.

## 3. Results

### 3.1. Descriptive Statistics and Correlation Analysis

Table 1 shows the descriptive statistics of the different variables analysed that were evaluated in the coaches (N = 57). Pearson’s correlation analysis showed, in both Table 1 and Table 2, correlations between variables in accordance with the theoretical bases. Negative relationships were shown, and most of them were significant for those variables related to fat percentage and visceral fat in relation to the rest of the constructs referring to physical fitness or the level of physical activity practised. Among the rest of the variables of physical fitness and sports practice, if significant, the relationships were positive.

Table 2 shows the descriptive statistics of the different variables analysed during the post-test. The correlations obtained are again in line with the theories of the field of physical activity and sport sciences that support the bidirectional relationships obtained between them.

### 3.2. Sample Normality Test

The Kolmogorov-Smirnov test revealed that all variables in this study had a normal distribution of data with significance *p* > 0.05.

### 3.3. Comparison of the Means between the Pre-Test and Post-Test

Table 3 shows the pre-test and post-test means for the different variables measured, in addition to the Student’s *t*-test with the degree of significance between the differences observed between one intake and the other, and the effect size with Cohen’s *d*, where small and medium effects were observed.

All the variables assessed showed significant changes, which in qualitative terms were positive in terms of improved body composition, physical condition and adherence to physical activity, as well as an increase in the number of days per week on which the participants practised some type of intense PA. 

A Wilcoxon test was performed for ordinal variables on the existence of significant changes in the perception of physical fitness and health. Table 4 shows the results of this test, with a significant increase in the perception of both constructs.

Figure 1 shows the percentages of the responses obtained between the pre-test and post-test, grouping them into good and very good on the one hand, and bad and very bad on the other hand. There was an increase in the responses for a perception of physical fitness (22.90–45.70%) and health (50.90–71.90%) between good and very good, and on the other hand, a decrease in the perceptions of considering physical fitness as bad and very bad (28.10–10.50%) and the same for health (1.80–1.80%).

## 4. Discussion

The aim of this research project was to analyse the effects of a physical exercise programme on physical activity levels, physical fitness, body composition and the perception of physical fitness and health in university students. After analysing the pre/post-intervention results, improvements were observed in all aspects of body composition evaluated after a 10-week training period, highlighting higher lean weight, lower fat percentage and lower visceral fat. These results are in line with previous research [47,48] where after a strength training intervention, lasting 8–10 weeks, respectively, participants obtained improvements in their body composition in terms of reduced body fat percentage and increased muscle mass. Following the hypotheses set out above, we could establish the achievement of H1. The improvement in body composition, in the previously mentioned terms of reduction of body fat and increase in muscle mass, leads to various proven improvements in health [49], and on the negative side, a loss of muscle mass would be associated with the development of chronic diseases, poorer quality of life, greater morbidity and health care admissions [50].

On the other hand, results related to H2 findings were significantly increased, indicating better physical fitness, flexibility, manual dynamometry, horizontal jump and core trunk strength. An intervention by Hammami et al. [51] shows how the implementation of programmes based on functional and strength-enhancing work reports an improvement in physical fitness variables such as upper and lower body strength (i.e., jumping, sprinting, manual press), as well as better general and specific dynamic coordination, among other factors. Therefore, the implementation of physical activity programmes, despite their short duration (10 weeks), produces quantifiable improvements in relation to variables related to physical fitness [52], with the ideal being their continuity and future adherence in order to obtain permanent and preventive improvements over time [53], as seen in this study where participants’ weekly PA practice days were increased, confirming H3 and revealing an indication of a higher consistency with exercise.

In addition, it was observed how participants had modified their self-perception of their physical fitness and health, improving both in a positive and significant sense, as stated in H4 and the last hypothesis of this research. These results have been confirmed in previous research where the practice of physical activity has had a positive impact on the improvement of self-perception of physical fitness and health in both young people [54] and older people [55]. Physical exercise is associated with a positive perception of health and fitness; thus, subjective and objective health variables are favoured [56], which will have beneficial consequences for those people who maintain an active lifestyle [57]. This is a compelling reason for the promotion of exercise-based interventions in the population that encourages continuity and adherence to healthy physical activity habits.

Participating in a physical activity programme consistently during the established period and observing improvements at a personal level influence participants to be more motivated to subsequently continue with physical exercise in an autonomous manner, which leads to higher levels of adherence to the practice of physical activity as observed in this study and in previous scientific literature [58]. 

Obviously, this study presents some limitations that should be underlined. First, there was a large disparity between the numbers of men and women included in the study, which might result in a rather heterogeneous sample. Moreover, as indicated in the methods section, the lack of a control group cannot fully infer the impact of an intervention. It would be necessary to supplement the study with a control group in future research.

The research prospects proposed would be to extend this study longitudinally over a longer period of time, which would make it possible to observe the possible improvements obtained in the programme and subsequently, by means of a retention period, to observe the maintenance or not of the improvements achieved after participation in the programme. In addition, prospective research should assess more aspects such as cardiorespiratory fitness for a more complete assessment of physical fitness. It also raises the possibility of further deepening the study by qualitatively evaluating the experiences of the coaches during their intervention using the SL methodology. Furthermore, it would be desirable to replicate this study considering different populations.

## 5. Conclusions

In conclusion, the findings of this study indicate that a 10-week training programme based on muscle strengthening, with a service-learning methodology, produced a significant improvement in the level of physical activity, in physical fitness-health variables in body composition and in the perception of physical fitness and health status of university students. 

## Figures and Tables

**Figure 1 ijerph-20-03377-f001:**
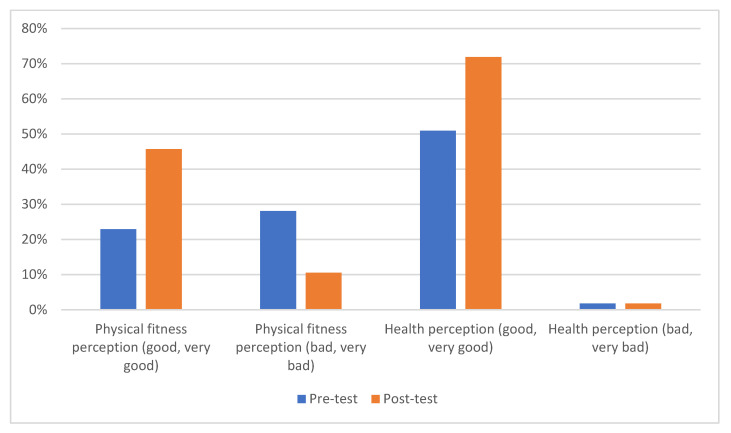
Percentage of responses for perception of physical fitness and health.

**Table 1 ijerph-20-03377-t001:** Descriptive Statistics and Correlations between Variables (Pre-test).

	*M*	*SD*	1	2	3	4	5	6	7	8	9
1. Lean W.	26.14%	5.60	-	−0.46 ***	−0.14	0.05	0.86 ***	0.86 ***	0.68 ***	0.45 ***	0.56 ***
2. Fat	28.65%	8.52		-	0.91 ***	−0.23	−0.46 ***	−0.50 ***	−0.72 ***	−0.51 ***	−0.30 *
3. Visc. Fat	7.93%	4.19			-	−0.22	−0.21	−0.23	−0.55 ***	−0.38 **	−0.09
4. Flex.	1.51 cm	7.29				-	−0.02	0.02	0.12	0.05	−0.02
5. Right H.P.	32.50 kg	9.78					-	0.94 ***	0.70 ***	0.55 ***	0.46 ***
6. Left H.P.	30.63 kg	8.83						-	0.72 ***	0.55 ***	0.47 ***
7. Jump	1.56 mts	0.37							-	0.56 ***	0.27 ***
8. Stabil.	01:04.70 min:seg	00:41.79								-	0.28 ***
9. I.P.A.	1.96 Likert	1.94									-

Note: W.: Weight; Visc.: Visceral; Flex: Flexibility; H.P.: Hand Pressure; Stabil: Central stability; I.P.A.: Intense Physical Activity. **** p* < 0.001; ** *p* < 0.01; * *p* < 0.05.

**Table 2 ijerph-20-03377-t002:** Descriptive Statistics and Correlations Between Variables (Post-test).

	*M*	*SD*	1	2	3	4	5	6	7	8	9
1. Lean W.	26.55%	5.57	-	−0.48 ***	−0.14	0.05	0.89 ***	0.87 ***	0.67 ***	0.25	0.39 **
2. Fat	27.17%	8.19		-	0.91 ***	−0.15	−0.53 ***	−0.50 ***	−0.67 ***	−0.45 ***	−0.25
3. Visc. Fat	7.35%	3.98			-	−0.14	−0.26 *	−0.24	−0.47 ***	−0.39 **	−0.11
4. Flex.	3.28 cm	6.73				-	−0.02	−0.02	0.06	−0.13	−0.022
5. Right H.P.	34.39 kg	9.75					-	0.92 ***	0.75 ***	0.41 **	0.31 *
6. Left H.P.	32.81 kg	9.47						-	0.77 ***	0.48 ***	0.31 *
7. Jump	1.61 mts	0.36							-	0.50 ***	0.13
8. Stabil.	01:28.13 min:seg	00:50.48								-	0.20
9. I.P.A.	3.16 Likert	1.42									-

Note: W.: Weight; Visc.: Visceral; Flex: Flexibility; H.P.: Hand Pressure; Stabil: Central stability; I.P.A.: Intense Physical Activity. **** p* < 0.001; ** *p* < 0.01; * *p* < 0.05.

**Table 3 ijerph-20-03377-t003:** Student’s *t*-test and effect size.

	*M1*	*M2*	*t*	CI (95%)	fd	*p*	*d*
1. Lean W.	26.14%	26.55 %	−3.621	(−0.643 to −0.185)	56	0.001	−0.074
2. Fat	28.65%	27.17 %	4.914	(0.881 to 2.094)	56	0.000	0.089
3. Visc. Fat	7.93%	7.35 %	3.858	(0.278 to 0.880)	56	0.000	0.142
4. Flex.	1.51 cm	3.28 cm	−3.845	(−2.699 to −0.849)	56	0.000	−0.252
5. Right H.P.	32.50 kg	34.39 kg	−3.842	(−2.872 to −0.903)	56	0.000	−0.193
6. Left H.P.	30.63 kg	32.81 kg	−4.412	(−3.179 to −1.193)	56	0.000	−0.239
7. Jump	1.56 mts	1.61 mts	−3.208	(−0.079 to −0.018)	56	0.002	−0.187
8. Stabil.	01:04.57 min:seg	01:28.13 min:seg	−5.072	(−32.86 to −14.25)	56	0.000	−0.508
9. I.P.A.	1.96 Likert	3.16 Likert	−6.289	(−1.573 to −0.813)	56	0.000	−0.706

Note. *M1*: Mean obtained from the pre-test; *M2*: Mean obtained from the post-test; W.: Weight; Visc.: Visceral; Flex: Flexibility; H.P.: Hand Pressure; Stabil: Central stability; I.P.A.: Intense Physical Activity.

**Table 4 ijerph-20-03377-t004:** Wilcoxon test.

	*M1*	*M2*	*z*	*p*
Physical fitness perception	2.95	3.37	−3.870	0.000
Health perception	3.65	3.89	−2.746	0.006

Note. *M1*: Mean obtained from the pre-test; *M2*: Mean obtained from the post-test.

## Data Availability

Data presented in this study are available on reasonable request from the first author.

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
