# Peer review of "Effects of a Physical Exercise Programme through Service-Learning Methodology on Physical Activity, Physical Fitness and Perception of Physical Fitness and Health in University Students from Spain: A Preliminary Study"

_ijerph, 2023, doi:10.3390/ijerph20043377_

Round 1
Reviewer 1 Report
The aim of this study is to analyse the effects of a physical exercise programme on levels of physical activity, physical fitness, body composition and the perception of physical fitness and health in university students using a methodology based on Service-Learning. The data reported highlight the importance of finding interventions to improve all these parameters in the university population.
This manuscript appears to be well-written, researched, and organized. I recommend accepting the paper after solving the following comments.
Abstract- In the summary it is mentioned that 18 students acted as coaches while in Participants it is said that there were 12.
Introduction- It would be appropriate to dedicate a brief description to the variables of interest (physical activity levels, physical fitness, body composition and the perception of physical fitness and health status). In addition, it could be justified why it is important to study these variables and not others in a university sample.
lines 68-70, In the aim of the study it is not clear that the methodology was based on Service-Learning
lines 87-90, Were the inclusion criteria the same for coaches as for students who received the program? If not, what were the inclusion criteria for them? I recommend clarifying it.
line 108, Remove missing parenthesis.
Results- Were the 12 students who acted as coach part of the participants evaluated pretest and posttest? I understand that no, but it is not clear in the manuscript. Could it be indicated that the results shown are from the other 57 students?
References- The cited references aren't mostly recent publications (within the last 5 years). Could this aspect be improved?
Reviewer 2 Report
The paper presents very interesting research about influence of Physical exercise to physical activity (PA), physical fitness (PF) and perception of PF and health in Spanisch University student. In my opinion, should be corrected (if possible) according to the comments below:
- Tittle: if all students are from Spain it should be included in the title.
- line 22: Mage should be corrected to M.
- line 37: citing unpublished studies is not a good practice, in my opinion this should be corrected.
- Line 47: The National Institute of Statics of which country? Please specify.
- The research is a pre-post experiment so I would suggest a research hypothesis.
- What is the reason for such a large disparity between the numbers of men and women?
- Line 141: Main variables is not good tittle of this subsection. This subsection does not discuss variables but devices and methods.
- The physical fitness tests used are very selective. With such a structured intervention, the use of only 4 fitness tests does not give the full impact of the intervention. For example, there is no test of cardiorespiratory endurance whose changes under the influence of the applied intervention would be very interesting.
- In statistical analysis, the authors mention the Student's test. What type of test was it for dependent or independent samples?
- In my opinion, the presentation of correlations between different measurements (tests) is not related to the purpose of the work. Please justify in the text the purpose of using correlation.
- In Table 3, the columns t, fd are redundant.
- Figure 1 should be replaced by a table (with N and %) with a chi-square test applied. The chi-square test would evaluate the relationship between the measurement and the PA perception and health perception categories.
- The discussion is very short, using only six citations. The first paragraph is essentially a restatement of the purpose and summary of the study. In my opinion, it should be significantly expanded.
- As the authors themselves note, one limitation is the lack of a control group. With pre-post experiments, a control group is essential. One cannot infer the impact of an intervention when there is no control group. It would be necessary to supplement the study with a control group, however, I do not know if this is still possible at this stage of the research.
Reviewer 3 Report
Dear authors,
Your work contributes to increase the importance of using different methodologies as Service-Learning at the university. The document is well-written and show the physical benefits related with perceptions. However, I have some considerations.
Methods:
- Which was the percentage of assistance to the program? Was there any percentage that was a exclusion criteria?
- Line 113: "return to bed". It is better cool-down.
- Description of body composition: Which were the conditions over the people were assessed? Same hour, without breakfast...
Results:
- For the table 1, 2 and 3 add the units of measurement of each variable.
Discussion:
- Along the discussion is only revealed the benefits from the population who participate in the study. But in this methodology, is also important the opinions from the trainers who give the service and also are learning. So, could you add any information about the results from this side?
Thank you.
